# RepFair-GAN: Mitigating Representation Bias in GANs Using Gradient Clipping

**Kamil Sabbagh[1], Patrik Joslin Kenfack[1], Adil Khan[1,3], & Adín Ramírez Rivera[2]**
[1]Innopolis University, [2]University of Oslo, [3]University of Hull
{k.sabbagh, p.kenfack}@innopolis.univeristy, a.khan@innopolis.ru
a.m.khan@hull.ac.uk, adinr@uio.no

## Abstract

This work introduces a new notion of fairness, *representational fairness*, for generative models, which ensures uniform representation of demographic groups in the generated data. Vanilla GANs violate this notion even when groups are equally represented. The proposed solution is to use *group-wise* gradient norm clipping to control gradient groups' magnitude during discriminator training. Experiments show that this method improves *representational fairness* while maintaining sample quality.

## 1 Introduction

The paper proposes a training process for GANs to generate data uniformly with respect to different groups at test time, addressing representation bias. We show that the magnitude of the groups' gradient on the discriminator side during the training plays an important role in guiding the generator to sample groups uniformly at the test time. Our process clips the gradient norm of groups to improve GANs' ability to uniformly generate samples from equal and unequal groups representation in the training data. Related work includes FairGAN Xu et al. (2018) that aims to generate samples that can achieve statistical parity for downstream classifiers, while Xu et al. (2019) considers other group fairness notions for classification.

## 2 Representational Fairness

Previous works Tan et al. (2020); Choi et al. (2020) have studied biases in GANs that generate non-uniformly distributed data over sensitive attributes. Balanced group representation doesn't prevent bias Kenfack et al. (2021); We hypothesize that the non-uniform generation results from disparities in gradient magnitude during discriminator training.

We consider a dataset $D = \{X, S\}$ with $N$ data points, where $\mathcal{X} = \{x_i\}_{i=1}^N$ represents samples from the distribution $P_{data}(X)$ and $\mathcal{S} = \{s_i\}_{i=1}^N$ represents the binary sensitive attribute for each point. A GAN estimates $P_\theta(X) = P_{data}(X)$ by generating realistic data $D_\theta = g_\theta(Z)$ through the generator $g_\theta : \mathbb{R}^d \to \mathbb{R}^{n \times n}$. Where $Z$ is a $d$ dimensional noise vector. Using function h : X $\to$ S, we can analyze the distribution of sensitive attributes in the generated data $P(h(g_\theta(Z)))$.

*Definition*: A well-trained generator $(g_\theta)$ is said to satisfy $\varepsilon$-*representationally fair* if it generates data, $D_\theta = g_\theta(Z)$, such that the distribution of the sensitive attribute $S$ is close to the uniform distribution, $\mathcal{U}(S)$, i.e., dist$(P(h(g_\theta(Z))), \mathcal{U}(S)) \leq \varepsilon$.

We aim to achieve *representational fairness* by enforcing the distribution of sensitive attributes in the generated data to be close to the uniform distribution, i.e., KL(P(h(g_\theta(Z))) | $\mathcal{U}(S)$) = $\varepsilon$. When $\varepsilon = 0$, the generative model achieves strict uniformity over the sensitive attributes.

## 3 EXPERIMENTS

First, we used SVHN Netzer et al. (2011) data set divided into two groups based on (synthetically) shaded and normal images as in Kenfack et al. (2021). we trained a vanilla GAN and a RepFairGan, figure 3 shows the gradient of each model. Then used a classifier with 98% accuracy to classify the generated images by each model. The images generated by the Vanilla GAN were 42.68% normal images and 57.32% shaded, for the RepFair Gan it was 49% normal and 51% shaded.

In the second experiment, we used MNIST LeCun et al. (1998) we divided it into two groups based on background colors black and white then we trained a CGAN Mirza & Osindero (2014) to generate all the digits of the MNIST data set while training on digits with both black and white backgrounds, Then we used a classifier with 100% accuracy to classify the number of the generated images of each background color for each digit. Figure 2 shows the results of the trained CGAN and RepFairCGAN. Figure 6 shows the results for different maximum gradient norms chosen.

all the models were trained on their respective datasets with different group representations. Experiments were based on 10 training runs and averaged across runs. Examples of generated images are shown in figures 5, 7, and 8.

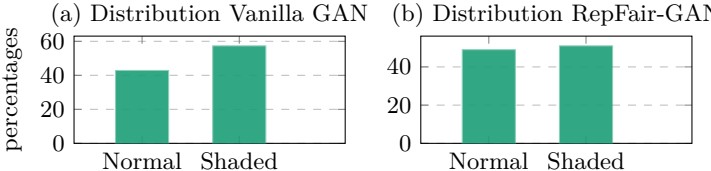

Figure 1: Group distributions for vanilla GAN (a) and our method (b) on SVHN dataset trained with balanced group representation. Our method provides uniform generation compared to the vanilla GAN.

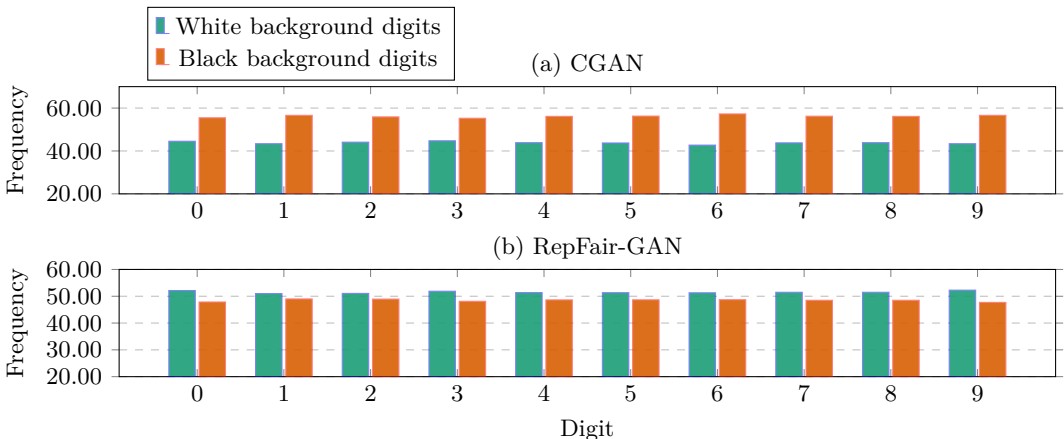

Figure 2: Class-wise background color distribution of CGAN outputs trained on MNIST dataset. (a) Imbalanced digit color distributions for each class produced by CGAN. (b) Balanced digit color distributions for each class produced by RepFair-GAN.

## 4 CONCLUSION

We proposed a new GAN training approach to address representation bias caused by gradient norm disparities during discriminator training. Empirical evaluation on MNIST and SVHN demonstrated significant improvement in group uniformity even in imbalanced datasets.

URM STATEMENT

The authors acknowledge that at least one key author of this work meets the URM criteria of ICLR 2023 Tiny Papers Track.

ACKNOWLEDGMENT

This work was supported by The Analytical Center for the Government of the Russian Federation (Agreement No. 70-2021-00143 dd. 01.11.2021, IGK 000000D730321P5Q0002)

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

## A    THE RELATION BETWEEN THE GRADIENT VECTORS AND REPRESENTATION BIAS

Fig. 3 shows that when there is a disparity in groups' gradient norms there is unfairness in the GANs generation, even when training data has an equal representation of both groups. We designed a training process (Algorithm 1) that controls the gradient of each group during training by modifying the discriminator training and enforcing a maximum gradient norm threshold to clip their gradients.

During CGAN training, gradient norms were tracked, and Fig. 4 shows that the gradient magnitude difference between groups widened over time in an unfair CGAN, while in the proposed method, which produced the results shown in Fig. 2. As the gradients become more proximate, the balance between different classes improves, highlighting the connection between gradient and class representation.

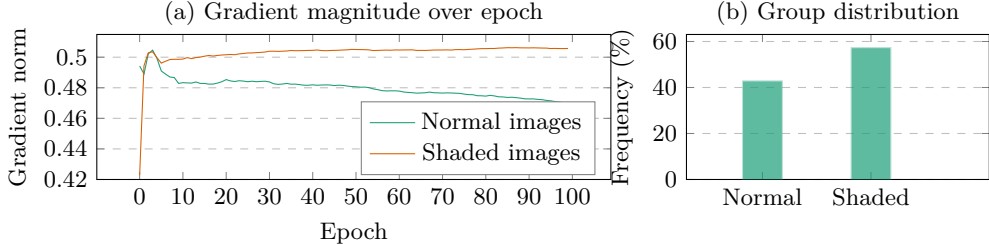

Figure 3: Unfair vanilla GANs on SVHN dataset had increasing differences in average gradient magnitudes between groups during training (a). After training, the generator sampled more from the group with a larger gradient (b).

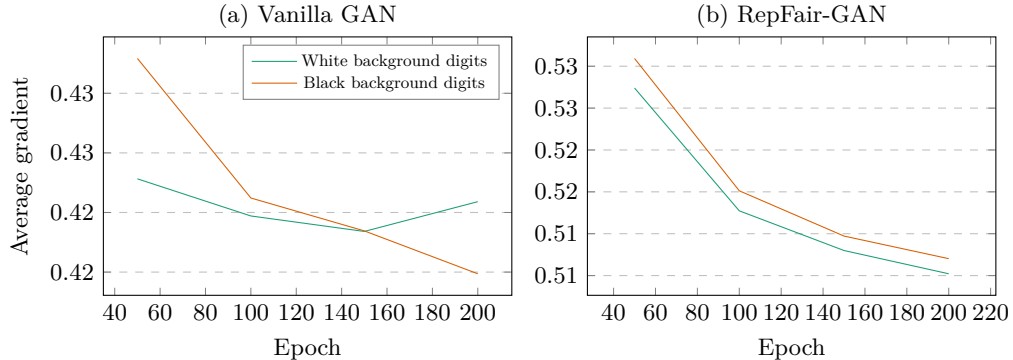

Figure 4: Gradient clipping was used to enforce closer gradient norms of each group during discriminator training, and its effect on the disparity in gradient norms was studied.

---

**Algorithm 1** Training of the discriminator in RepFair-GAN (outline)

---

**Require:** $t$ the batch size, $M$ training epochs, $C$ maximum gradient norm, and $\mathcal{L}_D(\Theta)$ the discriminator's loss function with network parameter $\Theta$.

   **for** $n \leftarrow 1, M$ **do**
      $U \leftarrow True$        $\triangleright$ Boolean indicating whether to update with samples from group 0 or group 1
      **for** each batch of size $t$ **do**
         **if** U **then**
            $X_t \leftarrow \{D|S = 0\}$                  $\triangleright$ random mini-batch of real samples from group 0
         **else**
            $X_t \leftarrow \{D|S = 1\}$                  $\triangleright$ random mini-batch of real samples from group 1
         **end if**
         $g_t \leftarrow \nabla_\Theta \mathcal{L}_D(\Theta; X_t)$           $\triangleright$ Compute the gradient using the discriminator's loss
         $\bar{g}_t \leftarrow \text{clip}(g_t, C)$               $\triangleright$ Clip the gradient norm with max norm $C$
         $\Theta \leftarrow \Theta - \alpha \bar{g}_t$                    $\triangleright$ Update network param.
         $U \leftarrow \text{not } U$
         Generate samples from the generator $\bar{X}_t$ feed to the discriminator and take a gradient step.
      **end for**
   **end for**

---

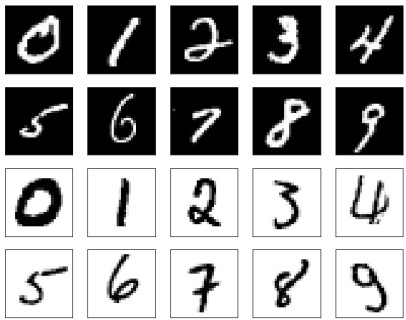
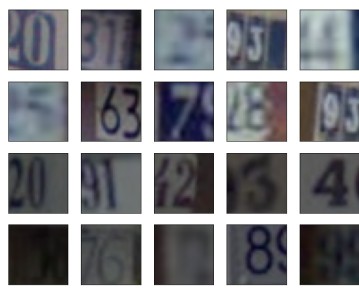

Figure 5: Example of two groups created from MNIST (left) and SVHN (right) datasets. In MNIST, one group of digits has a black background (two first rows) and another group has a white background (two last rows). In SVHN, one with normal images (first two rows), and for the second group we added a black overlay (shaded)

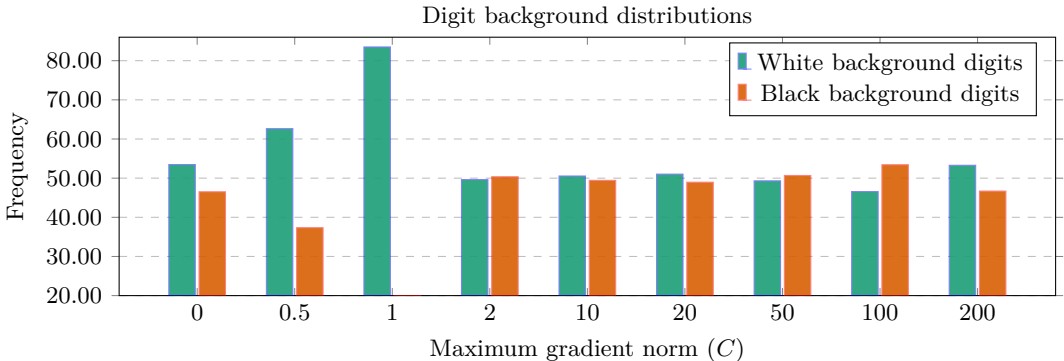

Figure 6: Group distributions with different maximum gradient norms in RepFair-GAN on MNIST data. For max norm less than 2 the training does not converge due to the vanishing gradient. For max gradient norms between 2 and 10 groups are sampled uniformly and the sampling becomes biased for values bigger than 20.

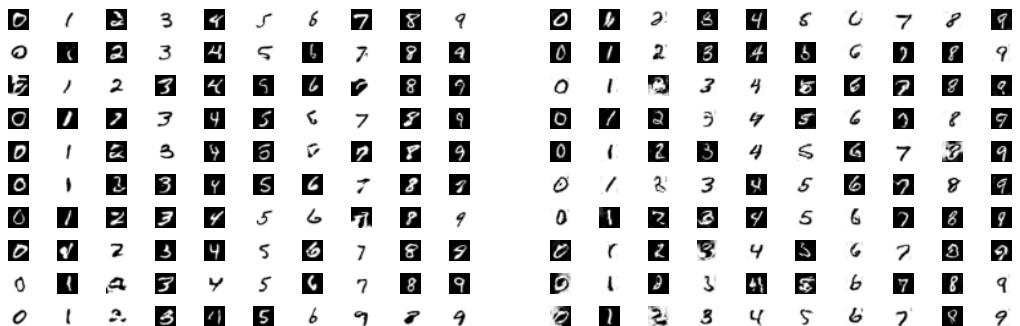

Figure 7: Qualitative results on MNIST dataset. Examples of digits generated by CGAN (left) and RepFair-GAN (right). We observe that for each type of digit, group are equally represented with our method while preserving the same quality as the classic GAN

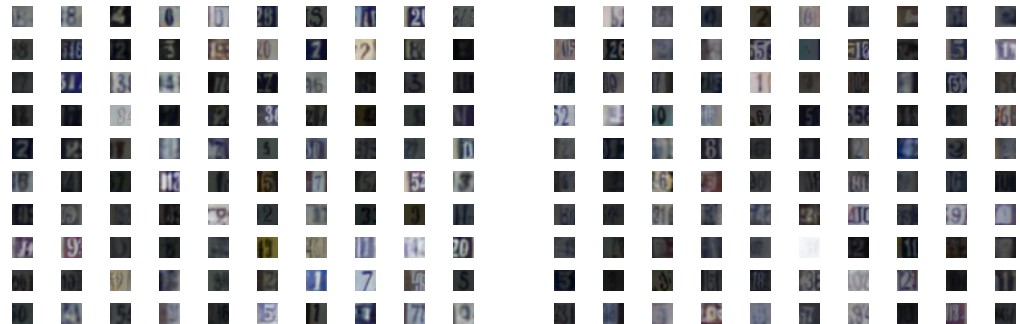

Figure 8: Qualitative results on SVHN dataset. Examples of images generated by GAN (left) and RepFair-GAN (right). We observe uniform generation of our method along with similar images quality

