# OpenReview forum: "RepFair-GAN: Mitigating Representation Bias in GANs Using Gradient Clipping"
_ICLR.cc/2023/TinyPapers — Submitted to Tiny Papers @ ICLR 2023_

### Official Review · Reviewer_mhR8 · 2023-03-24

**Confidence:** 3

**Summary Of Contributions:**

The paper proposes a new scheme for training GANs to mitigate representation bias through group-wise gradient norm clipping. The experiments show that the proposed method is effective in average gradient (fig. 2), frequency (fig. 3) and distributions (fig. 5 & 6).

**Rating:**

High Potential (HP): a submission which meets the reviewing criteria and has potential to make an impact on the field

**Strengths And Weaknesses:**

Strengths:
1. The topic that this work focuses on is interesting and the proposed RepFair-GAN seems effective in that regard.
2. The idea of RepFair-GAN is illustrated clearly with well-written descriptions and experiments.

Weakness:
1. Although claimed to be task-agnostic, experiments of downstream tasks performance comparisons between RepFair-GAN and vanilla GANs would be meaningful to demonstrate the usefulness of RepFair-GAN.

**Suggested Changes:**

Although the author(s) argue that the proposed work does not target a specific task for classification, it should be understood as such scheme can improve multiple downstream tasks. Therefore, it would be still meaningful to conduct further experiments and analysis on if RepFair-GAN would help achieve better task results than other baseline GANs.

---

> ### Author Response · Authors · 2023-05-11
> **Addressing Reviewer Feedback: responding to issues with the inclusion of downstream task performance comparisons, presenting experimental setup, and demonstrate the improvements in fairness**
>
> Dear Reviewer mhR8,
>
> Thank you for your constructive feedback and recognition of the interesting topic and clarity of our proposed RepFair-GAN. We appreciate your time and effort in reviewing our work.
>
> In response to your suggestion regarding the inclusion of downstream task performance comparisons, we have revised our paper to address this concern. We initially mentioned two experiments in our work: one with vanilla GAN and another with CGAN. However, we acknowledge that our initial presentation of the results and experimental setup may not have been clear enough.
>
> To address your feedback, we have provided a more detailed explanation of the experimental setup and the GAN models used. Furthermore, we have added the corresponding figures to demonstrate the improvements in fairness and the effectiveness of our method. We believe these revisions will help to illustrate the usefulness and potential impact of RepFair-GAN in various downstream tasks.
>
> We hope these changes address your concerns and enhance the overall quality of our paper.

---

### Official Review · Reviewer_yNZA · 2023-04-02

**Confidence:** 4

**Summary Of Contributions:**

This paper proposed achieving representational fairness by using group-wise gradient norm clipping to control gradient groups’ magnitude during discriminator training

**Rating:**

Great Start (GS): a submission which meets some of the reviewing criteria but has room for improvement

**Strengths And Weaknesses:**

**Weaknesses**
1. The datasets used in this paper is not for fairness purpose. I recommend using datasets for fairness.
2. The experiment only explores the behavior of the gradient, which does not report the fairness performance.

**Strengths**
1. Analyzing the behavior of the gradient for fairness is interesting and important.


**Suggested Changes:**

1. Add datasets for fairness
2. Evaluate the fairness performance.

---

> ### Author Response · Authors · 2023-05-11
> **Addressing Reviewer Feedback: responding to issues with not using fairness-specific datasets, and the not having performance evaluation metrics.**
>
> Dear Reviewer yNZA,
>
> Thank you for your feedback and constructive comments on our paper. We appreciate your recognition of the importance of analyzing gradient behavior for fairness.
>
> In response to your concerns regarding fairness datasets, we acknowledge that our current experiments were primarily designed as a proof of concept. Due to computational constraints, we could not run our experiments on fairness-specific datasets such as FairFace. However, we fully intend to expand our experiments in the future to test our claims on such datasets, which we believe will further strengthen our work.
>
> Regarding your comment about the lack of fairness performance evaluation, we have made revisions to address this issue. We have moved certain sections from the appendix to the main text, which demonstrate the improvements in representational fairness achieved by our method. Concurrently, we have relocated the detailed explanation of gradient behavior to the appendix. By incorporating the necessary figures into the main text and moving the gradient figures to the appendix, we believe our paper now presents a clearer and more comprehensive overview of our work's contributions.
>
> We hope these revisions address your concerns and enhance the overall quality of our paper.

---

### Comment · Area_Chair_8dfL · 2023-06-01
**This work meets the threshold for archival, contains the URM statement and is deanonymized**

---

### Meta-Review · Area_Chair_8dfL · 2023-04-08

**Recommendation:** Invite to revise
**Confidence:** 5

**Metareview:**

The authors operationalize representational fairness for image generation and argue that gradient magnitudes may be driving the imbalance in generation even when the training data is balanced. The main issue that reviewers point out with clarity is that many of the key results (i.e. that the proposed method actually increases fairness over existing methods) are left in the appendix. As a result, the main text is more difficult to follow and the results in the main text are hard to interpret. The reviewers also suggest that reproducibility and correctness of results would be strengthened if the authors could demonstrate the results on additional datasets.

**Summary:**

The authors argue that representational fairness can be improved by clipping gradients but the reviewers point out that most of the crucial evidence for this claim is in the appendix.

**Comments And Feedback To The Authors:**

Please consider shortening the introduction and bringing in some of the keys results from the appendix into the main text, including extending discussion of these results. This would help your proposed method really shine by providing all the necessary evidence for supporting your main claim.

**Reason For Not Giving A Higher Recommendation:**

This is an interesting result but clarity could be greatly improved with the suggested changes.

**Reason For Not Giving A Lower Recommendation:**

N/A

---

> ### Author Response · Authors · 2023-05-11
> **Revised Paper: Addressing Reviewer Feedback for Improved Clarity and Presentation**
>
> Dear Reviewer and Area Chair 8dfL,
>
> Thank you for your valuable feedback and insightful suggestions on our paper. We appreciate the time and effort you have put into reviewing our work. In response to your comments, we have made several revisions to improve the clarity and presentation of our paper. Specifically, we have shortened the introduction to ensure that it remains concise and focused on our main contributions. Additionally, we have moved the key results from the appendix to the main text, providing a more comprehensive and easy-to-follow presentation of our findings. Lastly, we have expanded the discussion of our results in the main text, highlighting the improvements in representational fairness achieved through our proposed method. We believe these changes address your concerns and strengthen the quality of our paper.  we wish to opt-in for archival of the paper.

---

### Decision · Program_Chairs · 2023-04-10

Revision accepted; invite to archive